# The Effect of the Static Load in the UNSM Process on the Corrosion Properties of Alloy 600

**DOI:** 10.3390/ma12193165

**Published:** 2019-09-27

**Authors:** Ki Tae Kim, Young Sik Kim

**Affiliations:** Research Center for Energy and Clean Technology, School of Materials Science and Engineering, Andong National University, 1375 Gyeongdong-ro, Andong, Gyeongbuk 36729, Korea; kitae1@pyunji.andong.ac.kr

**Keywords:** alloy 600, UNSM, corrosion, static load, residual stress

## Abstract

To suppress stress corrosion-cracking, compressive residual stresses, such as shot peening, laser peening, water jet peening, ultrasonic peening, and ultrasonic nanocrystal surface modification (UNSM) are utilized. However, among the numerous techniques, there is little research about the corrosion effect of detailed conditions, such as static load or amplitude in UNSM. A study on UNSM among various techniques of adding compressive residual stress to Alloy 600 was conducted. The focus of this study was on the effect of the static load in UNSM on the corrosion properties of Alloy 600. Microstructure analysis was conducted using an optical microscope (OM), a scanning electron microscope (SEM), and electron backscattering diffraction (EBSD), while compressive residual stress was measured using a nano-indentation technique. A cyclic polarization test and the AC (Alternating Current)-impedance measurement were both used to analyze the corrosion properties. An increase in static load under critical static load enhanced the grain boundary diffusion, consequently strengthened the passive film, and facilitated the surface diffusion, thereby improving the passivation of Alloy 600. However, higher static loads over the critical value can lead to an increase in the friction between the striking tip and the surface, thereby creating an overlapped wave, which reduces the corrosion properties.

## 1. Introduction

Ni alloy is among the fcc (face centered cubic) crystal structures; it has good strength, ductility, resistance to corrosion, and oxidation resistance at high temperatures [1]. Therefore, it is commonly used as a nuclear power plant material because of its high temperature corrosion and high temperature strength. Alloy 600 and Alloy 690 are used in nuclear reactors and steam generators in the nuclear power plants’ primary system [2,3]. There are reports that Alloy 600 is susceptible to primary water stress corrosion cracking in field applications [4,5,6,7], hence Alloy 690 has been used for new plants or as a replacement material for the Alloy 600. However, Alloy 600 is used in majority of the plants. SCC (stress corrosion cracking) constitutes the major damage due to Alloy 600, and the damaged areas include the CMDR (control rod drive mechanism) nozzle of the reactor head, the BMI (bottom mounted instrumentation) nozzle of the reactor bottom, a dissimilar metal weld of the RPV (reactor pressure vessel) nozzle, and the heat transfer tube of the steam generator [8].

There are many techniques to mitigate the stress corrosion cracking of Alloy 600, which include substitution with high corrosion resistant materials, modification of water chemistry, and reduction of residual stress. Among these remedies, the reduction of residual stress, including tensile stress and compressive stress, can be achieved through surface treatment. Techniques used for adding compressive residual stresses to materials include: shot peening [9,10,11], laser peening [12,13,14,15,16], water jet peening [17,18], ultrasonic peening [19,20], and ultrasonic nanocrystal surface modification (UNSM), with many studies still ongoing. Water jet peening and laser peening have been used to mitigate SCC, since 2001 and 2004, respectively. In the USA, nuclear power plants, such as Byron-2, Braidwood-1, Wolf Creek and ANO-1 in 2016; and Callaway, Byron-1, Braidwood-2, and ANO-2 in 2017–2018 used laser peening, and further research has been conducted in this regard [2].

Regarding the UNSM process, the material is impacted with a hard, rigid pin moving at an ultrasonic frequency, typically 20 kHz. A tungsten carbide (WC) tip is then attached to an ultrasonic horn, which strikes the specimen surface at about 20,000 times per second with 1000 to 10,000 shots made per square millimeter within a very short time. This impact deforms the surface of the target material and converts its microstructure into nanocrystals. The plastic deformation produces a 1–2 mm deep compressive residual stress field in the material. The combination of surface deformation, the nanostructured layer, and compressive residual stresses causes a delay in fatigue crack initiation [21].

The UNSM process consists of various variables, including static load, amplitude, pitch, tip diameter, speed, etc. Numerous studies have been performed using limited conditions among UNSM process. The reported effects of UNSM on the mechanical properties and microstructure can be summarized as follows [22,23,24,25,26]: (i) The microstructure of the surface and under surface is converted to a nanograined microstructure which enhanced yield strength, tensile strength, and hardness as a result of the Hall–Pitch strengthening mechanism. (ii) Very high levels of compressive residual stresses were formed (about (1–2 GPa). High compressive residual stress enhanced the resistance to abrasion, wear, and fatigue lifetime. (iii) A new phase was formed—in an austenitic stainless steel, martensite can be formed through a UNSM process and the formed martensite phase in turn protects the formation of the necking, since the new phase increases the interface strength between the inside matrix and the nanograined surface. (iv) Surface roughness can also be improved. On the other hand, UNSM has influenced the corrosion properties as follows [27,28,29,30,31]: (i) Since the UNSM process strengthens the passive film, the pitting corrosion resistance of austenitic stainless steels can be enhanced. (ii) UNSM treatment of aged, austenitic stainless steel reduced chromium carbide and carbon segregation, hence improving the inter-granular corrosion resistance. (iii) Introduction of high compressive residual stress enhanced the resistance to SCC. Additionally, many studies about the UNSM’s effect on Ti alloys and carbon steel have been performed. However, there is little research on the detailed conditions, such as static load or amplitude, among UNSM conditions.

Therefore, this work focused on the effect of the static load to the surface via UNSM on the corrosion properties of Alloy 600. The effects of the static load on the corrosion properties were discussed on the basis of the microstructure, hardness, and residual stress.

## 2. Materials and Methods

### 2.1. Specimen

Alloy 600 was utilized for this study. Table 1 shows the chemical composition of the experimental alloy. Heat treatment was applied to the specimen at 1040 °C for 7 min. UNSM treatment was applied to the surface using the UNSM equipment (Design Mecha-LM20 UNSM system, Asan, Korea). The UNSM process is schematically presented in Figure 1 and UNSM treatment conditions are summarized as in Table 2. The surface preceding the UNSM treatment was ground with #2000 SiC abrasive paper. As illustrated in Figure 1, UNSM treatment was applied in a zig-zag manner and 2.38 mm tungsten carbide tip was used with an amplitude of 30 μm and a pitch of 0.07 mm, as summarized in Table 2. This work controlled the static load—10 N, 30 N, and 50 N.

### 2.2. Microstructure Analysis

The microstructure was observed through optical microscopy (AXIOTECH 100HD, ZEISS, Oberkochen, Germany), and the surface profile was obtained with the use of a 3D microscopy (KH-7700, HiROX, Tokyo, Japan). Furthermore, the surface shape was observed by SEM (VEGA II LMU, Tescan, Brno, Czech Republic) after the UNSM process. Microstructural properties were analyzed using EBSD (Electron Backscattering Diffraction, Lyra 3 XMH, Oxford Ins., Abingdon, UK). Specimens for EBSD analysis were ground using #4000 SiC paper, and then polished using a colloidal silica (0.04 μm). The EBSD step size was 0.3 μm and the observed data were post-processed using the HKL channel 5 (Oxford Ins., Abingdon, UK) analysis software. The crystallographic structure and lattice plane spacing were analyzed by XRD (X-ray diffraction) (Ultima IV, Rigaku, Tokyo, Japan) and the scan rate was 1°/min.

### 2.3. Hardness and Residual Stress Measurements

The samples’ hardnesses were measured using a micro-Vickers hardness tester (HV-100, Mitutoyo, kawasakishi, Japan). Residual stress was determined using a nanoindentation system (Inforce 1000, KLA, Milpitas, USA) with a Berkovich indenter with a maximum load of 100 mN and a holding time of 10 s. An average of 10 measurements were used for each reported data point.

### 2.4. Electrochemical Tests

The corrosion test was performed in a solution of 1% NaCl at 30 °C using a potentiostat (interface 1000, Gamry, Warminster, VA, USA). The saturation calomel electrode (SCE) was used as the reference electrode, while a high-density graphite rod was utilized as the counter electrode. De-aeration was done using N_2_ gas at the rate of 100 mL/min for 30 min.

#### 2.4.1. Cyclic Polarization Test

The specimen was coated with epoxy apart from a distance 1 cm^2^ after epoxy mounting. In the cyclic polarization test, the forward and reverse scan rates were both 0.33 mV/s. Additionally, the cyclic polarization apex current density was 0.1 mA/cm^2^. In the forward polarization, passive current density and pitting potential were obtained, and the protection potential was defined as the intercept in the reverse and forward polarization curves.

#### 2.4.2. AC-Impedance Measurement

The test specimen was similar to that of the polarization test. Before measuring, passivation was treated at +0 mV (SCE) for 30 min. AC impedance was measured at a passivation potential of between 10 kHz to 0.01 Hz, and the AC voltage amplitude was 10 mV.

## 3. Results

### 3.1. The Effect of the Static Load in UNSM Treatment on the Microstructure of Alloy 600

UNSM treatment causes a change in various properties of materials. Figure 2 shows the effect of the static load in UNSM treatment on the appearance of Alloy 600 through an optical microscope (magnification: ×50). The static load was varied between 10 N, 30 N, and 50 N. The surface appearances were similar to each other regardless of the static load; many micro valleys were observed. These valleys were associated to the applied load and the pitch of UNSM treatment. Figure 3 shows the 3D microscope observation. Figure 3 also reveals the effect of the static load in UNSM treatment on the surface topography of Alloy 600 through a 3D microscope (magnification: ×600). An increase in the static load results in a deeper valley-depth as follows: 9.0 μm for 10 N, 11.1 μm for 30 N, and 11.2 μm for 50 N specimens. It suggested that valley depth may be associated with the intrinsic mechanical properties, such as yield strength and elastic modulus [21].

The cross-section of the microstructure by UNSM treatment was observed through an EBSD. Figure 4 shows the effect of the static load in UNSM treatment on the grain refinement depth profile of Alloy 600 by EBSD. Untreated specimen showed a typical austenitic microstructure, including twins and large grains (Figure 3a). Regardless of the static load, grain refinement proceeds from the surface into the matrix. In the case of the 10 N static load, the depth of grain refinement was about 40 μm; the depth was 80 μm for the 30 N static load, and about 90 μm for the 50 N static load. As shown in the figures below, the density of grain boundary was increased with an increase in the UNSM static load resulting in a deeper effective depth. Figure 5 summarizes the effect of the static load in UNSM treatment on the grain diameter distribution of Alloy 600. There are a lot of large grain diameters in the untreated specimen. By increasing the static load in UNSM treatment, the distribution function of smaller grain diameter was increased. Notably, the UNSM treatment could refine the grain size in addition to increasing the static load, amount of grain refinement, and the refined depth.

The surface layer subjected to plastic deformation presents a high-density dislocation through striking and produces a very small lamellae deformation-twin (DT). However, due to constant blows, the dislocation density is high, creating shear bands (SB) and cutting the DT/DT intersection. The DT and SB distort the microstructure and change into grains that are equiaxed and refined by the development of the newly formed grain boundaries, thereby reducing the grain size [32,33,34].

As described above, UNSM treatment changed the microstructure from the surface into the matrix. Figure 6 shows the effect of the static load in UNSM treatment on the hardness depth profile of Alloy 600. The outermost surface revealed the maximum hardness, higher static load, and higher surface hardness. The hardness depth profiles reduced greatly, irrespective of the static load, and the hardness-depth relationship showed a polynomial expression. Although a higher static load resulted in a little higher of a hardness depth profile, similar hardness attenuation was revealed as follows; y means the hardness (HV) and x implies the depth (μm) from top surface. As shown in below, the surface hardness increases with an increase in the static load. However, the hardness depth profile of higher static load specimen decreases more compared to that of lower static load specimen. Finally, the effect of the static load on hardness depth profile diminished from the depth of about 500 μm.

Figure 7 shows the effect of the static load in UNSM treatment on the dislocation density of Alloy 600 through kernel average misorientation (KAM). Dislocation densities can be obtained using a KAM map. In the KAM map, the red > yellow > green color series suggests a higher dislocation density qualitatively. As shown in Figure 6, an increase in the static load, increased the dislocation density resulting in an increased depth.

However, it is interesting that the depth of grain refinement is different to the hardness depth profile, as the depth of high dislocation density is different with the hardness depth profile. This difference can be explained as follows; a sharp increase in the hardness of the outer surface was as a result of grain refinement and the increased dislocation density, as shown in Figure 5 and Figure 7 (it is noteworthy that the detection depth by EBSD was about 120 μm). The grain refinement through UNSM treatment was limited to the outer surface, but the increased hardened depth was deeper than the depth of grain refinement via a work hardening mechanism.

Since UNSM treatment involves hitting the surface numerous times using a strong WC tip in a very short period, high energy can be applied to the surface of materials, leading to a transformation. Figure 8 shows the effect of the static load in UNSM treatment on the XRD patterns of Alloy 600. Phase transformation were not detected in XRD analysis, but an increase in the static load increased the half width of peak. The increased half width of the peak suggests the formation of amorphous state [35], and it is considered that this may be related to the grain refinement using UNSM treatment.

### 3.2. The Effect of the Static Load in UNSM Treatment on the Residual Stress of Alloy 600.

It was confirmed that UNSM treatment resulted in grain refinement, the formation of an amorphous solid, increased dislocation density, and high hardness. To understand how these microstructural properties affect the residual stress, the residual stresses were determined using a nanoindentation technique [36,37]. Figure 9 shows the effect of the static load in UNSM treatment on the representative load-displacement curves of Alloy 600 using Nano-indentation. Untreated specimen suggests that the stress-free-state curves of UNSM-treated specimen were located more left than those of the treated stress-free state, which demonstrates the presence of compressive residual stress [38]. Regardless of the static load, the compressive residual stress was higher at the outer surface, as shown in figures below.

The residual stress calculated from Figure 9 and Figure 10 summarized the effect of the static load in UNSM treatment on the residual stress depth profile of Alloy 600. High compressive residual stress was obtained through an increase in the static load and the stress was reduced with depth, as shown in Figure 10.

Formation of the compressive residual stress through UNSM treatment can be explained as follows: the peening of the surface the material was from cyclic plastic loading, due to progressive peening effects. Outer layers experience an in-plane stretching plastic deformation, whereas the elastic sub-surface attempts to maintain its original shape, hence generating compressive residual stress at the surface [39,40]. Additionally, a small portion of the applied stress is stored in the form of residual stresses within the structure as a tangled network of dislocations [41]. A very fine grain with high dislocation density as a result of UNSM treatment could facilitate the formation of compressive residual stress.

### 3.3. The Effect of the Static Load in UNSM Treatment on Corrosion Properties

Figure 11 shows the effect of the static load in UNSM treatment on the polarization curve of Alloy 600 in a deaerated solution of 1% NaCl at 30 °C. Regardless of the static load, the specimens display the trans-passive transition. Table 3 summarizes the corrosion factors (E_pitting_: pitting potential, E_protection_: protection potential, and i_passive_: passive current density) obtained from Figure 11 due to the static load in UNSM treatment. Regardless of the static load value, the pitting potential was increased, and the passive current density was decreased compared to the untreated specimen, but the higher static load decreased the protection potential. This behavior implies that the static load enhances resistance to corrosion through UNSM treatment, as indicated by the critical value. A static load greater than the critical value may be harmful to corrosion resistance.

Figure 12 shows the effect of the static load in UNSM treatment on the AC impedance measurement for the passive film formed at 0 V (SCE) in a deaerated solution of 1% NaCl at 30 °C. Figure 12a represents the Nyquist plot, and Figure 12b plotted the polarization resistance obtained from Figure 12a using Randel’s model. UNSM treatment enhanced the polarization resistance regardless of the static loading, until the critical value where the resistance started to decrease. This behavior was similar to the polarization test results.

## 4. Discussion

As described above, UNSM treatment of Alloy 600 changed its mechanical properties, microstructural properties, and corrosion resistance, increasing the static load. Surface hardness was greatly increased, and the hardness depth profile was reduced from the surface to the matrix. UNSM treatment could refine the grain size, and moreover, the higher the static load, the greater the amount of grain refinement and the refined depth were increased. When the static load increases, high compressive residual stress is caused, and the stress reduces with depth. Regardless of the static load, UNSM treatment enhanced the resistance to corrosion, but decreased when the load exceeded the critical value. Why did the static load in UNSM treatment enhance the corrosion resistance? Why was its effect dependent upon the static load values?

Figure 13 illustrates the effect of the static load under the critical UNSM condition on the corrosion factors of Alloy 600. Specifically, under the critical static load, increasing the static load improved the pitting potential and protection potential, and it reduced the passive current density compared to untreated specimen.

This study sought to understand the enhancement of corrosion resistance in regard to grain refinement and residual stress. Corrosion behavior through the reduction of grain size is dependent on the alloy type, corrosion environment, and corrosion forms [42]. Claims that improved corrosion resistance is as a result of grain refinement are generally attributed to an improved passive film [43,44,45,46]. Grain boundaries contain higher energies than the bulk, and as such [47] are more chemically active, hence a high density of grain boundaries increases the reactivity at the surface due to increased electron activity and diffusion. In addition to enhanced rates of diffusion, the main consequence of sites such as grain boundaries and triple junctions, include reduced atom coordination and increased electron activity [48]. Increased reactivity, coupled with more sites for the nucleation of an oxide film on the surface of grain-refined materials is suggested, to result in a more rapid formation of a protective layer. In the case of austenitic stainless steels and Ni-alloys, it was reported that Nano-crystallization improved the corrosion resistance [45,49,50,51,52].

Figure 14 shows the effect of the static load in UNSM treatment on lattice plane spacing of Alloy 600. Figure 14a shows the lattice plane spacing via the static load where an increase in the load reduced the lattice plane spacing. From this result, the decrement model of lattice plane spacing using UNSM treatment was suggested as shown in Figure 14b, and the driving force for the decrement model is the compressive residual stress induced through UNSM treatment.

Based on the results of O. Takakuwa [53], corrosion resistance is improved when compressive residual stress is induced through cavitation peening; this might be the reason why the reduction of inter-atomic spacing due to the compressive stress at the surface facilitates the growth and maintenance of the passivation film. The compressive residual stress enhances not only the mechanical properties, but also the corrosion resistance. Therefore, it can be concluded that UNSM treatment refines the grain size and this facilitates the grain boundary diffusion, thereby strengthening the passive film, and its treatment induces compressive residual stress which facilitates the surface diffusion, and subsequently improves the passivation of Alloy 600.

However, a static load exceeding the critical value can degrade the corrosion resistance, as shown in Figure 15. Figure 15 illustrates the effect of the static load in the critical UNSM condition on the corrosion factors of Alloy 600. Through increasing the static load, the protection potential was decreased as the passive current density increased, although the pitting potential was increased. Specifically, the reduction of corrosion resistance through UNSM treatment may be as a result of another factor.

Figure 16 illustrates the effect of the static load on the surface morphologies and mechanical overlapping via UNSM treatment. As shown in the optical micrographs, UNSM treatment led to valley pitch by pitch, and many lines, like cracks, were also observed. Mechanically overlapped lines were observed in SEM images. B. Wu [54] reported that the surface can be damaged by increasing the friction between the striking tip and surface due to high static load during UNSM process.

From the OM and SEM observations, the mechanical overlapping model of UNSM treatment was proposed as shown Figure 17; in the initial stage (Figure 17a), WC tip strikes the surface. When the tip travels along the moving direction, many waves are formed (Figure 17b). Through the repeated process, the overlapped area, which acts as the initiate site of corrosion, could be formed. This overlapped wave formation may reduce the corrosion resistance; and increasing the static load could accelerate corrosion and decrease the protection potential.

## 5. Conclusions

This work was centered on the effect of the static load in UNSM treatment on the corrosion properties. The conclusions are as follows:

(1) Regardless of the magnitude of the static load, UNSM treatment enhanced the corrosion resistance, until it decreased when the load exceeded the critical value. Increasing the static load under the critical static load enhanced the pitting potential and protection potential, and reduced the passive current density compared to untreated samples. It can be concluded that UNSM treatment can refine the grain size which facilitates the grain boundary diffusion, and therefore, strengthen the passive film; and its treatment could induce the compressive residual stress, which enhances the surface diffusion and subsequently improves the passivation of Alloy 600.

(2) However, a high static load exceeding the critical value can degrade the corrosion resistance. After increasing the static load, the protection potential was reduced and the passive current density was increased. The surface may be damaged by increasing the friction between the strike tip and surface due to the high static load during the UNSM process. Through that repeated process, an overlapped wave, which acts as the initiating site of corrosion, may be formed. This overlapped wave formation may reduce the corrosion resistance.

## Figures and Tables

**Figure 1 materials-12-03165-f001:**
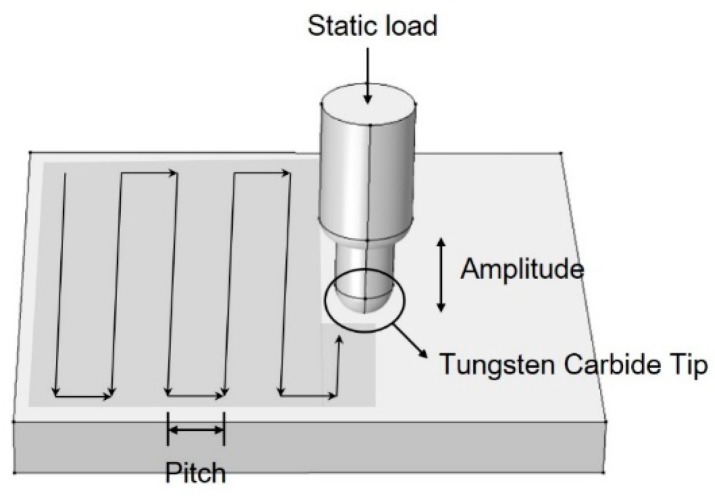
Schematic of the ultrasonic nanocrystal surface modification (UNSM) process.

**Figure 2 materials-12-03165-f002:**
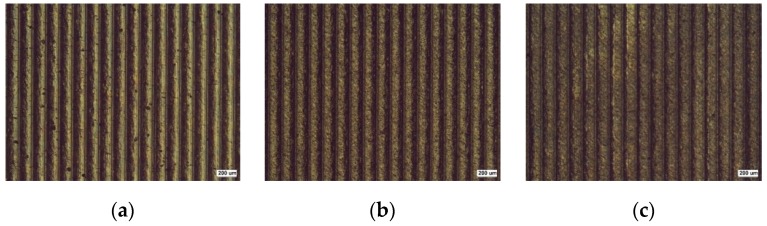
The effect of the static load in UNSM treatment on the appearance of Alloy 600 by optical microscope (×50): (**a**) 10 N; (**b**) 30 N; (**c**) 50 N.

**Figure 3 materials-12-03165-f003:**
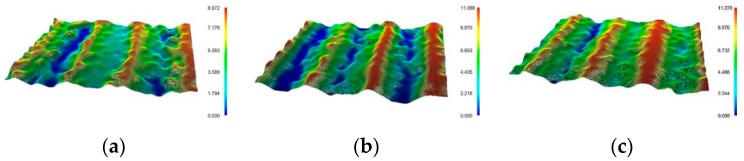
The effect of the static load in UNSM treatment on the surface topography of Alloy 600 by 3D microscope (×600): (**a**) 10 N; (**b**) 30 N; (**c**) 50 N.

**Figure 4 materials-12-03165-f004:**
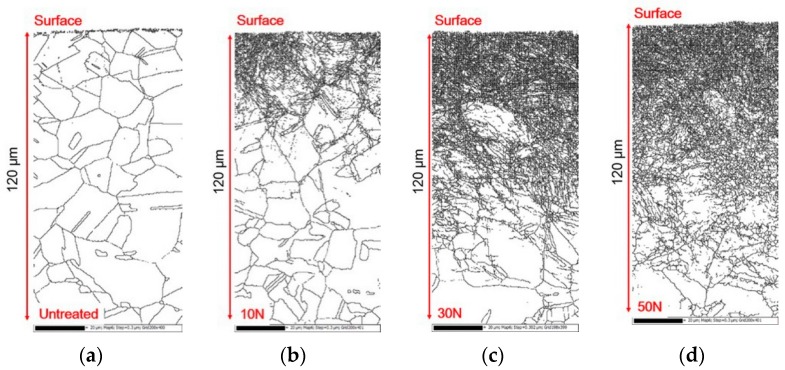
The effect of the static load in UNSM treatment on the grain refinement depth profile of Alloy 600 by EBSD (×600): (**a**) untreated; (**b**) 10 N; (**c**) 30 N; (**d**) 50 N.

**Figure 5 materials-12-03165-f005:**
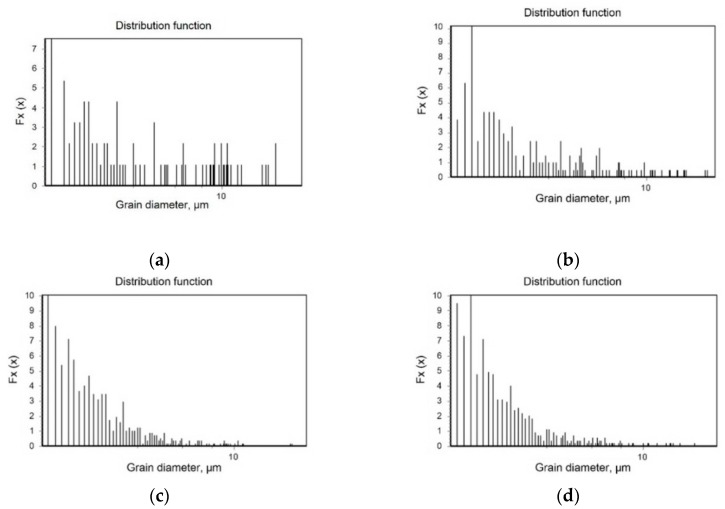
The effect of the static load in UNSM treatment on the grain diameter distribution of Alloy 600 determined from the EBSD area of Figure 4: (**a**) untreated; (**b**) 10 N; (**c**) 30 N; (**d**) 50 N.

**Figure 6 materials-12-03165-f006:**
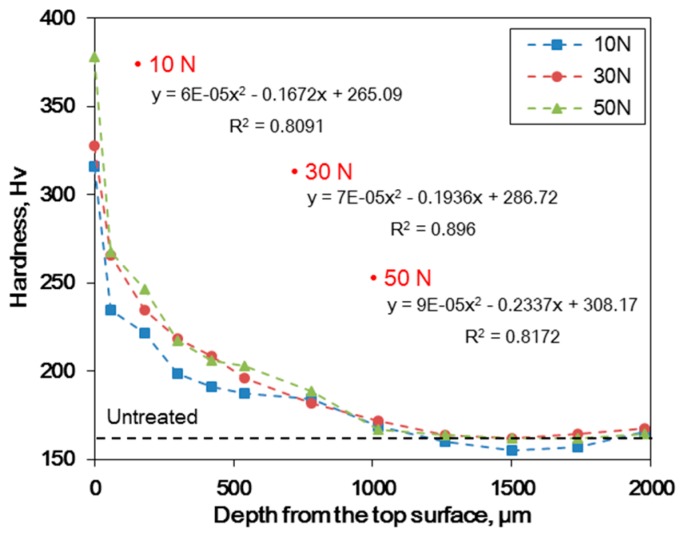
The effect of the static load in UNSM treatment on the hardness depth profile of Alloy 600.

**Figure 7 materials-12-03165-f007:**
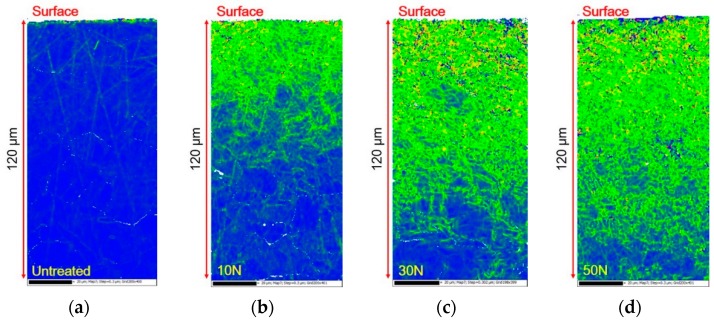
The effect of the static load in UNSM treatment on the dislocation density of Alloy 600 by kernel average misorientation; (**a**) untreated; (**b**) 10 N; (**c**) 30 N; (**d**) 50 N.

**Figure 8 materials-12-03165-f008:**
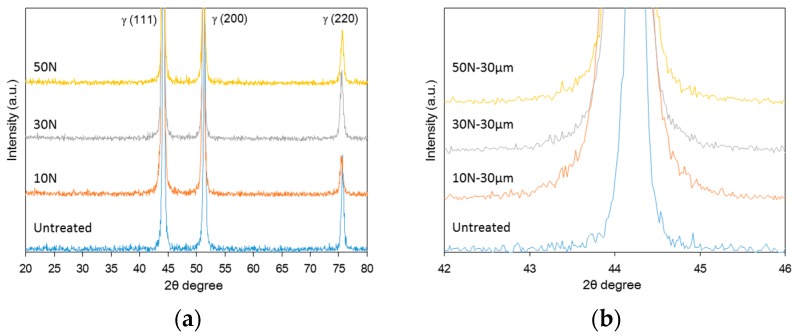
The effect of the static load in UNSM treatment on the XRD patterns of Alloy 600: (**a**) 20°–80°; (**b**) 42°–45°.

**Figure 9 materials-12-03165-f009:**
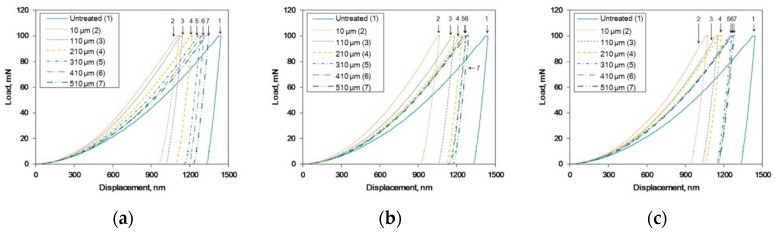
The effect of the static load in UNSM treatment on the representative load-displacement curves of Alloy 600 by nano-indentation: (**a**) 10 N; (**b**) 30 N; (**c**) 50 N.

**Figure 10 materials-12-03165-f010:**
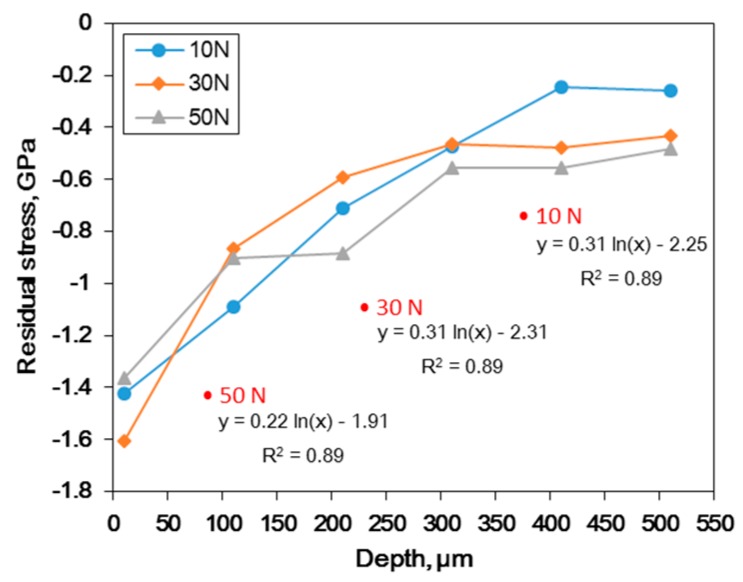
The effect of the static load in UNSM treatment on the residual stress depth profile of Alloy 600.

**Figure 11 materials-12-03165-f011:**
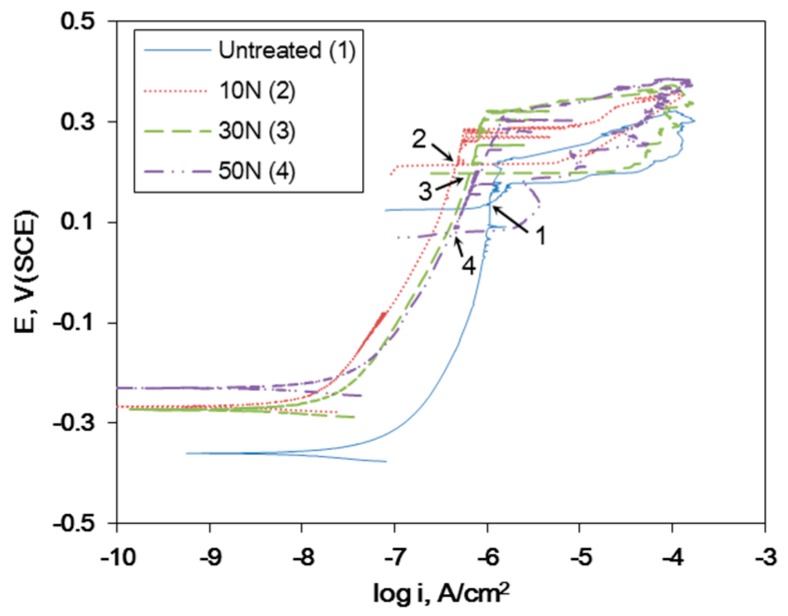
The effect of the static load in UNSM treatment on the polarization curve of Alloy 600 in deaerated 1% NaCl at 30 °C.

**Figure 12 materials-12-03165-f012:**
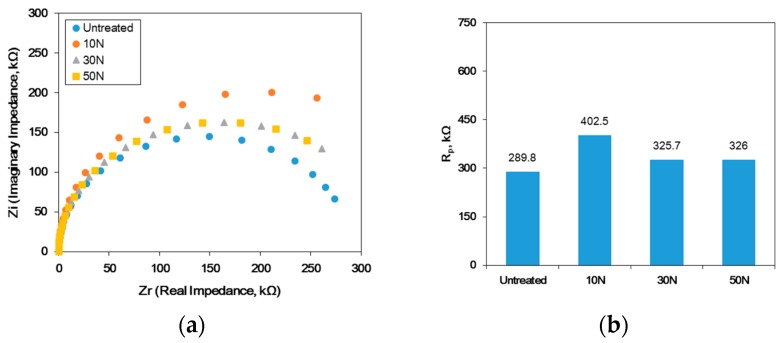
The effect of the static load in UNSM treatment on the AC impedance measurement for the passive film formed at 0 V (saturation camel electrode, SCE) in deaerated 1% NaCl at 30 °C: (**a**) Nyquist plot and (**b**) polarization resistance.

**Figure 13 materials-12-03165-f013:**
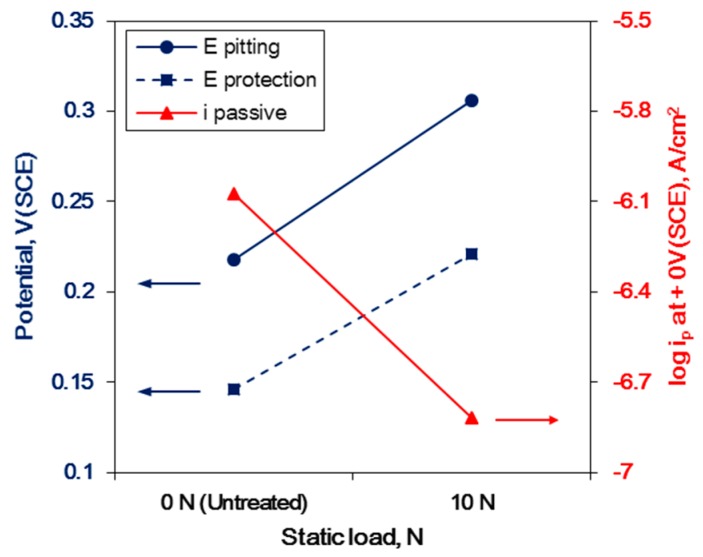
The effect of the static load under the critical UNSM condition on the corrosion factors of Alloy 600.

**Figure 14 materials-12-03165-f014:**
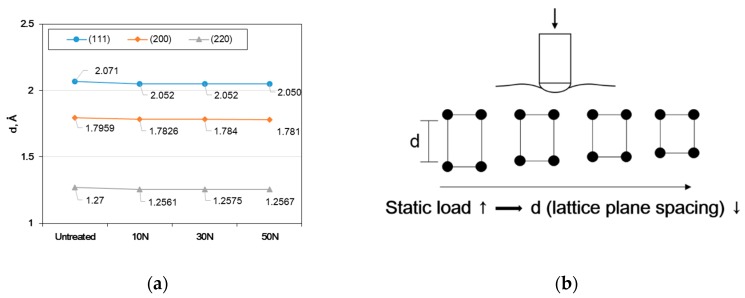
The effect of the static load in UNSM treatment on lattice plane spacing of Alloy 600; (**a**) measured lattice plane spacing and (**b**) decrement model of lattice plane spacing by UNSM treatment.

**Figure 15 materials-12-03165-f015:**
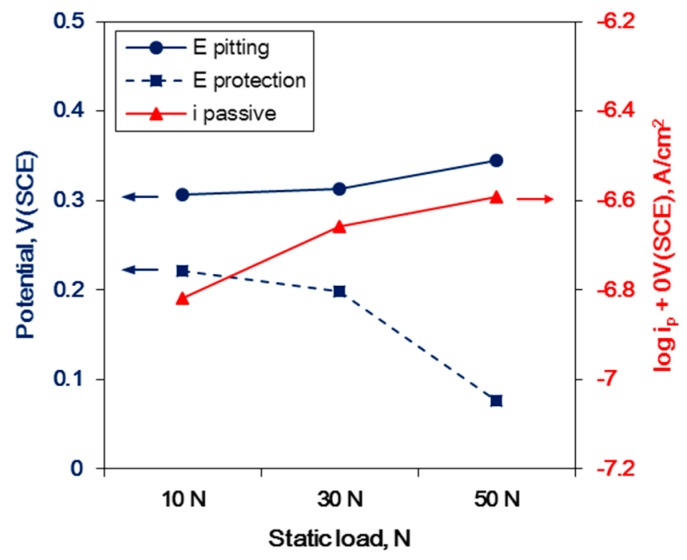
The effect of the static load over the critical UNSM condition on the corrosion factors of Alloy 600.

**Figure 16 materials-12-03165-f016:**
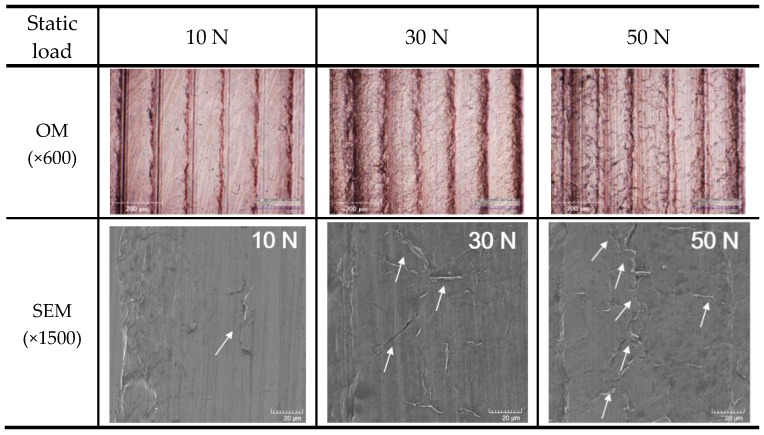
The effect of the static load on the surface morphologies and mechanical overlapping by UNSM treatment.

**Figure 17 materials-12-03165-f017:**
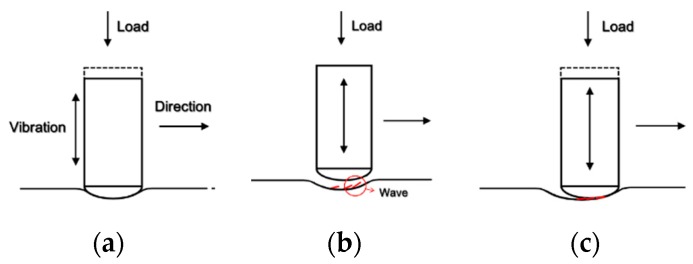
Mechanical overlapping model by UNSM treatment: (**a**) beginning stage; (**b**) wave formation; (**c**) ultrasonic tip movement; (**d**) overlapped area formation and the evidentiary image.

**Table 1 materials-12-03165-t001:** Chemical composition of Alloy 600 used in the experiments (wt. %). (This chemical composition is provided by the manufacturer).

Ni	Cr	Fe	C	Mn	S	Si	Cu
73.13	16.35	9.42	0.07	0.21	0.002	0.29	0.01

**Table 2 materials-12-03165-t002:** UNSM conditions for Alloy 600.

Sample Name	Static Load (N)	Amplitude (μm)	Pitch (mm)	Tip Diameter(mm)
Untreated	-	-	-	-
10 N	10	30	0.07	2.38 (WC^1^)
30 N	30	30	0.07	2.38 (WC)
50 N	50	30	0.07	2.38 (WC)

^1^ WC: Tungsten Carbide.

**Table 3 materials-12-03165-t003:** Corrosion factors obtained from Figure 10 by the static load in UNSM treatment.

Static Load	Untreated	10 N	30 N	50 N
E_pitting_, V(SCE)	0.218	0.306	0.313	0.345
E_protection_, V(SCE)	0.146	0.221	0.198	0.079
i_passive_, A/cm^2^ at 0 V(SCE)	10^-6.074^	10^-6.819^	10^-6.658^	10^-6.593^

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
