# Peer review of "The Effect of the Static Load in the UNSM Process on the Corrosion Properties of Alloy 600"

_materials, 2019, doi:10.3390/ma12193165_

Round 1
Reviewer 1 Report
The paper”Effect of the Static load in UNSM process on the 2 Corrosion Properties of Alloy 600” is suitable for publication in your journal.
Author Response
Thank you for your comment.
Reviewer 2 Report
In my opinion, the article is very interesting. I suggest making some corrections.
Section 2. Materials and Methods
Rows 80-81
Was the chemical composition determined on the basis of measurements or is it the composition given by the manufacturer? It should be written.
Section 2.2 Microstructure Analysis
A description and working parameters of the scanning electron microscope applied should be given i.e. acceleration voltage, kind of image (SEI or BEI), the type of equipment.
XRD - the type of anode should be given. In addition, the authors can specify the type of equipment and software they have used.
Section 3. Results
Row 155
Whether this distribution function of grain diameter is measured from a specific area, e.g. according to Fig. 3. This should be described.
Figure 5
The standard deviation should be marked on the curves.
Row 253-256
There is in the article: “In the case of 10 N and 30 N, the pitting potentials and protection potentials were higher while the passive current densities were lower compared to those of the untreated specimen. However, in the case of 50 N, the pitting potentials were higher, and the passive current densities were lower compared those of untreated specimen.”
The first and second sentences sound the same. The sentences should be rewritten.
Reviewer 3 Report
The authors reported about the passive current, which would mean that the current is not changing with potential, but in fact it does - Figure 10.
That needs to be rephrased.
Reviewer 4 Report
This paper address the effect of static load on the mechanical properties of an alloy and its relationship with the corrosion resistance. I think with the following minor revisions this paper should be ready for publication.
(1) The discussion section needs a careful language editing.
(2) labels and lines in Fig.8 are hard to see. The authors should provide a clearer/more visible images for Fig. 8.
(3) The author stated the importance of critical value to enhance the polarization resistance. However, there is no clear statement on what is the particular value that can be assigned as the critical value. Since the author only examined the effect of 10, 30 and 50 N static load, it seems that 10 N is the critical value. I think if the authors could provide corrosion resistance data for 5 and 20 N, the readers could get a better understanding.
